# Long Noncoding RNA H19: A Novel Oncogene in Liver Cancer

**DOI:** 10.3390/ncrna9020019

**Published:** 2023-03-09

**Authors:** Yanyan Wang, Jing Zeng, Weidong Chen, Jiangao Fan, Phillip B. Hylemon, Huiping Zhou

**Affiliations:** 1Department of Microbiology and Immunology, Medical College of Virginia, Central Virginia Veterans Healthcare System, Virginia Commonwealth University, 1220 East Broad Street, MMRB-5044, Richmond, VA 23298, USA; 2School of Pharmacy, Anhui University of Chinese Medicine, Hefei 230012, China; 3Department of Gastroenterology, Xinhua Hospital, Shanghai Jiao Tong University School of Medicine, Shanghai 200092, China

**Keywords:** lncRNAH19, HCC, CCA, bile acids

## Abstract

Liver cancer is the second leading cause of cancer-related death globally, with limited treatment options. Recent studies have demonstrated the critical role of long noncoding RNAs (lncRNAs) in the pathogenesis of liver cancers. Of note, mounting evidence has shown that lncRNA H19, an endogenous noncoding single-stranded RNA, functions as an oncogene in the development and progression of liver cancer, including hepatocellular carcinoma (HCC) and cholangiocarcinoma (CCA), the two most prevalent primary liver tumors in adults. H19 can affect many critical biological processes, including the cell proliferation, apoptosis, invasion, and metastasis of liver cancer by its function on epigenetic modification, H19/miR-675 axis, miRNAs sponge, drug resistance, and its regulation of downstream pathways. In this review, we will focus on the most relevant molecular mechanisms of action and regulation of H19 in the development and pathophysiology of HCC and CCA. This review aims to provide valuable perspectives and translational applications of H19 as a potential diagnostic marker and therapeutic target for liver cancer disease.

## 1. Introduction

Liver cancer is a common malignancy worldwide; its incidence and mortality rates remain high and continue to increase. Due to the increase and aging of the worldwide population, as well as the increase in risky lifestyle behaviors, such as smoking, poor diet, and lack of exercise, liver cancer has become one of the six most common worldwide cancers [1,2]. Primary liver cancer mainly comprises two pathophysiological subtypes, hepatocellular carcinoma (HCC) and cholangiocarcinoma (CCA), with an incidence of 90% of HCC and 10% of CCA [3,4]. The pathophysiological processes of live cancers are very complex. Although current treatment methods, such as surgery, radiotherapy, and chemotherapy, have developed rapidly, the outcome is unsatisfactory due to low efficacy and highly adverse side effects [5]. In addition, the high mortality and morbidity associated with HCC and CCA have imposed substantial financial burdens on individuals and healthcare systems [6]. Therefore, there is an unmet need to develop new treatment options to improve the clinical outcomes of patients with liver cancer. In recent years, much progress has been made in elucidating the molecular mechanisms of hepatocarcinogenesis and development.

Long noncoding RNAs (lncRNAs) refer to a group of transcripts of more than 200 nucleotides in length but do not code genes for proteins in eukaryotic cells. LncRNA H19 was the first LncRNA and imprinted gene identified in eukaryotes in the late 1980s [7]. During the last three decades, extensive studies have shown that lncRNAs are involved in early cell development, proliferation, differentiation, apoptosis, and other metabolic processes [8]. The dysregulation of lncRNA expression has been associated with many human diseases, including cancers and metabolic disorders. H19 is highly expressed in fetal tissue but suppressed in adult tissues except for skeletal muscle. The abnormal expression of H19 is associated with various human cancers, including liver, gastric, pancreatic, and colorectal [9,10,11,12,13]. Recent studies reported that H19 plays a critical role in the progression of HCC and CCA, the two major pathophysiological subtypes of primary liver cancer. This review will summarize the current understanding of lncRNAs and H19 in hepatic cancer development. Furthermore, we will review the recent progress on the role of H19 in the pathogenesis of HCC and CCA and discuss its potential as a biomarker and target for therapeutic intervention.

## 2. LncRNAs and H19

### 2.1. LncRNAs

The sequencing of the entire human genome in the early 2000s established that the majority of the human genome (~93%) is transcribed, but less than 2% of transcripts encode proteins, and most transcripts represent noncoding RNAs (ncRNAs). NcRNAs are further classified according to their lengths. Short ncRNAs consist of highly abundant and functionally essential RNAs such as microRNAs (miRNAs), small interfering RNAs (siRNAs), ribosomal RNAs (rRNAs), transfer RNAs (tRNAs), and other types of small RNAs less than 200 nucleotides in length. LncRNAs are defined as RNAs whose transcripts exceed 200 bp. Similar to mRNAs, lncRNAs have exons and introns and are transcribed by RNA polymerase II; 5′ capping, 3′ polyadenylation, and splicing [14]. In addition, lncRNAs are tissue- and cell type-specific and can undergo epigenetic modifications and transcriptional regulation [15].

LncRNAs are classified into sense, antisense, bidirectional, intronic, and intergenic lncRNAs based on the topographic relationship with the nearest protein-encoding genes [16]. Recent studies indicated that lncRNAs are involved not only in early cell development, cell proliferation, differentiation, and apoptosis but also in many metabolic processes. Although numerous lncRNAs have been reported to be involved in liver cancer metastasis via genome-wide transcriptome analysis, their characterization and biological functions remain largely unknown compared to protein-encoding genes and short ncRNAs [17,18,19].

### 2.2. H19

H19 was the first discovered lncRNA which was cloned and characterized in the pre-genomic era. H19 lacks a typical open reading frame. The H19 locus belongs to a conserved gene cluster on chromosome 7 in mice and on chromosome 11p15.5 in humans. It contains five exons and four short introns, which results in a 2.3 kb transcript after fully capping, splicing, and polyadenylation [20,21]. H19 is highly expressed in embryonic tissues and the placenta, and its expression is markedly reduced after birth in most tissues except skeletal, cartilage, and cardiac muscle. In the liver, H19 is abundantly expressed at the fetal stage and significantly reduced in healthy adult livers. However, the expression of hepatic H19 is reactivated under pathological conditions, such as nonalcoholic steatohepatitis, cholestatic liver fibrosis, and liver cancers [22,23]. The relative expression levels of H19 in different types of hepatic cells depend on the pathophysiological conditions.

H19 was first identified as an imprinted gene. Genetic and molecular studies showed that H19 is a paternally imprinted and maternally expressed gene that lies downstream of insulin-like growth factor (IGF2), another maternally imprinted and paternally expressed protein-encoding gene [24]. Both H19 and IGF2 genes are expressed in the same tissues, and their reciprocal expression is coordinately controlled by the zinc-finger protein CCCTC binding factor (CTCF), which binds to an unmethylated maternal imprinting control region (ICR) and prevents the activation of IGF2 by enhancers located downstream of H19 [25,26]. The regulation of H19 and IGF2 expression was illustrated in a previous review [27]. It was also reported that the primary precursor of microRNA-675 (miR-675) is H19, which is embedded in the first exon of the H19 gene. The excision process of miR-675 from H19 is regulated by the RNA-binding protein (RBP) human antigen R (HuR) [28]. H19 acts as an independent lncRNA to regulate the expression level of miR-675. H19 can regulate various biological processes through the H19/miR-675 axis by targeting oncogenic or tumor-suppressive factors, as miR-675 has numerous targets across multiple signaling pathways [29]. In addition, H19 can also interact with other miRNAs, acting as a sponge and blocking the ability of these miRNAs to interact with their targets [30,31,32]. Collectively, the mechanisms by which H19 regulates cellular functions include epigenetic regulation, miR-675 production, miRNAs sponge action, and the regulation of target gene expression through binding to RBPs, among others [33,34].

## 3. LncRNA H19 in HCC

HCC is one of the leading causes of cancer-related mortality worldwide and is often diagnosed at late stages due to the lack of reliable diagnostic and prognostic biomarkers [35]. Currently, no effective treatment is available for HCC, and liver transplantation remains the only curative option. The aberrant upregulation of H19 in tumorigenesis has been well-documented in different types of human cancers [36], but the role of H19 in HCC appears more complex. A comprehensive review of H19 in HCC over the past three decades is reported elsewhere [37]. The most recent studies related to H19 in HCC are summarized in Table 1. Studies with cultured HCC cell lines, *in vivo* animal models, and human HCC samples indicated that H19 functions as an oncogene by regulating epigenetic modification, H19/miR-675 axis, miRNA sponge, drug resistance, and diverse signaling pathways. However, more mechanistic and comprehensive studies are needed to define the role of H19 in the progression of HCC.

### 3.1. Genetic Polymorphisms

Genetic polymorphisms affect the susceptibility and clinical outcomes of HCC [59]. Recent studies have focused on the association of single nucleotide polymorphisms (SNPs) in the H19 gene with the risk and prognosis of cancer [60]. Mingli Yang et al. found that an intron rs2839698 SNP of H19 was associated with an increased risk of HCC and has the potential to be a predictor for HCC risk and prognosis [61]. In contrast, Lili Ge et al. reported that the T allele of H19 rs217727 polymorphism apparently increased the survival rate of patients with HCC. H19 upregulated the expression of miR-675 and reduced the expression of FADD, caspase-3, and caspase-8. The rs217727 polymorphism in H19 promotes cell apoptosis by regulating the expressions of H19 and the activation of the miR-675/FADD/caspase-8/caspase-3/apoptosis signaling pathway [39]. The mutations of rs2839698 and rs3741219 of H19 conferred an increased susceptibility to HCC [62]. However, an association between SNPs, rs2107425 and rs3024270, and the occurrence and progression of HCC were observed [62]. Consistently, it has been reported that the rs3741219 SNP of H19 led to the allelic binding of miR-146b-3p/miR-1539, resulting in the allelic expression of H19 in HCC cells [63]. Taken together, the identification of individual H19 polymorphism is critical for understanding the pathogenic role of H19 in HCC and developing novel therapeutics for HCC. Table 2 summarizes studies that assessed the contribution of H19 SNPs to the risk of HCC.

### 3.2. Epigenetic Modification

Epigenetic modifications, characterized by DNA methylation, histone modifications, and chromatin remodeling, are important regulators of gene expression in many diseases, including HCC [64,65]. NSUN2 is an RNA methyltransferase responsible for the 5-methylcytosine (m5C) modification of multiple RNAs [66]. Aberrant NSUN2-mediated m5C modification of H19 promoted the occurrence and development of HCC by recruiting the Ras GTPase-activating protein-binding protein 1 (G3BP1), which is a DNA-unwinding enzyme and a key component of stress granules [46]. H19 was also reported to induce P-glycoprotein expression and MDR1-associated drug resistance in R-HepG2 cells by modulating MDR1 promoter methylation [67]. IGF2 is clustered with H19 on chromosome 7 in mice and controlled by an imprinting control region (ICR) located upstream of H19, a differentially methylated region. As expected, the methylation level decreased by nearly 30% in tumors compared to non-tumors, which correlates with elevated H19 levels [50]. The expression levels of IGF2 and H19 were significantly upregulated in HCC, mainly due to the decreased DNA methylation of the IGF2/H19 locus [50]. These results indicate that epigenetic modification is important for regulating H19 expression and its cellular/molecular functions.

### 3.3. H19/miR-675 Axis

MiR-675 is embedded in the first exon of H19. Many studies have shown that H19 could act as an independent lncRNA to regulate the expression of miR-675, which plays an oncogenic role in liver cancer. H19 can regulate various biological processes through the H19/miR-675 axis by targeting oncogenic or tumor-suppressive factors, as miR-675 has numerous targets and regulates various signaling pathways [29]. Yiqing Liu et al. reported that the H19/miR-675/PPARα axis regulated liver cell injury, energy metabolism, and remodeling induced by the hepatitis B X protein, which may be related to the modulation of the Akt/mTOR signaling pathway [42]. H19/miR-675 axis was reported to downregulate the expression levels of p53 [68,69]. Notably, the knockdown of H19 and miR-675 was shown to induce the expression of p53, eventually promoting cell apoptosis in human HCC cell lines [51]. Moreover, it has been reported that knockdown of H19 sensitized HCC cells to sorafenib by downregulating miR-675, thereby preventing EMT in HCC, as illustrated by increased E-cadherin and decreased vimentin expression [48]. In contrast, another study reported that the miR-675 expression level in the HCC liver tissues was lower than in healthy liver tissues, and the overexpression of miR-675 inhibited the growth of HCC-derived cell lines and promoted cell death, suggesting tumor suppressor activity of miR-675 [53]. Furthermore, this study demonstrated that miR-675 repressed the Fas-associated protein with death domain (FADD), resulting in cell necrosis and inflammation [53]. Interestingly, it was also reported that miR-675 could upregulate H19 through activating EGR1 in human liver cancer, suggesting a potential positive feedback loop of H19-miR-675 expression in HCC [70]. In addition, another study reported that the expression level of miR-675 reflected tumor dynamics more accurately than H19 and may represent a new biomarker for the diagnosis, prognosis, and monitoring of therapeutic responses in HCC [71]. The most recent understanding of the H19/miR-675 axis in HCC is illustrated in Figure 1. These results indicate that multiple mechanisms are involved in the H19-mediated regulation of HCC through H19/miR-675 axis.

### 3.4. Sponge of miRNAs

Recently, the competing endogenous RNA (ceRNA) hypothesis suggested that lncRNAs might function as molecular sponges for miRNAs in various cancers [72,73]. H19 can also act as a ceRNA by antagonizing miRNAs [74]. As a significant risk factor for hepatocarcinogenesis, chronic hepatitis B virus (HBV) infection is implicated in HCC development [75,76]. H19 was highly expressed in HBV-related HCC tissues and was found to promote the malignant development of HBV-related HCC by regulating miR-22 as a molecular sponge [41]. Another study by Wei L et al. reported that H19 could serve as a ceRNA to sponge miR-326 and modulate the expression of the transcription factor TWIST1 in HCC pathogenesis [43]. H19 knockdown inhibited the proliferation, migration, and invasion and promoted apoptosis of HCC cells via targeting the miR-15b/CDC42/PAK1 axis [45].

Extracellular vesicles (EVs) are membrane-released vesicles that can act as transporters for proteins, lipids, miRNAs, and lncRNAs [77]. A study measuring miRNA and the lncRNA transcriptional expression and secretion of EVs in an HCC cell line (HepG2) and a non-tumor hepatocyte cell line indicated that HepG2-derived EVs contained lower levels of H19 and different miRNAs compared to the non-tumor cell line, indicating possible different modes of regulation between normal and cancer cells [78]. EVs consist of exosomes, microvesicles, and apoptotic bodies [79]. Dongmei Wang et al. reported that exosomal H19 promoted proliferation, migration, and invasion and inhibited the apoptosis of HCC cells treated with Propofol through upregulating LIMK1 via sponging miR-520a-3p [47]. It has been reported that activated tumor-associated macrophages (TAMs) induced H19 expression and promoted HCC aggressiveness [49]. H19 functions as a sponge of miR-193b and triggers the activation of MAPK1 signaling pathways [49]. Bone is the second most frequent site of metastasis for HCC, which leads to an extremely poor prognosis [80]. HCC bone metastasis is typically osteolytic, involving the activation of osteoclasts. Zhao Huang et al. reported that H19 could promote HCC bone metastasis by upregulating zinc finger E-box binding homeobox 1 (ZEB1) via functioning as a sponge for miR-200b-3p [54]. The major role of H19 as miRNA sponges in HCC is summarized in Figure 2. The identification of H19 as miRNA sponges has significantly contributed to our understanding of the role of H19 in the progression of HCC.

### 3.5. Drug Resistance

Chemotherapy is one of the most important treatments for HCC, especially for people who have lost options for surgery and have not responded to local therapies [81]. However, drug resistance frequently arises and has become a major cause of cancer treatment failure. Studies have shown that H19 plays a key role in multiple drug resistance, especially in liver cancer, breast cancer, and colorectal cancer [82,83,84]. Yongzi Xu et al. reported that the knockdown of H19 elevated sorafenib sensitivity by suppressing EMT in HCC cells [48]. H19 was also reported to induce P-glycoprotein expression and MDR1-associated drug resistance in R-HepG2 cells by regulating MDR1 promoter methylation [67]. A previous study also demonstrated that H19/miR-193a-3p axis could modify the radio-resistance and chemotherapeutic tolerance of HCC cells by targeting Presenilin-1 (PSEN1), a core component of γ-secretase [85]. In contrast, the downregulation of H19 reversed the chemotherapy resistance of CD133 + cancer stem cells by blocking the MAPK/ERK signaling pathway in HCC [38]. In general, H19 can promote the resistance of HCC to chemotherapeutic drugs, and the mechanism of H19 in the chemotherapy resistance of HCC may provide more effective chemotherapy for drug-resistant HCC patients.

### 3.6. Cancer Stem Cells

It is widely accepted that both differentiated hepatocytes and cells with progenitor characteristics, termed cancer stem cells (CSCs), can lead to HCC [86,87]. Rojas Ángela et al. reported that H19 expression was increased in CSCs of liver tissue and plasma of HCC patients and decreased after partial/complete therapeutic response. Patients who developed HCC during the follow-up showed higher levels of H19 [88]. CD90+ liver CSCs display aggressive and metastatic phenotypes. Alice Conigliaro et al. reported that CD90+ liver CSCs modulate endothelial cell phenotype by releasing H19-containing exosomes [89]. It also has been reported that the downregulation of H19 could induce oxidative stress and reverse chemotherapy resistance of CD133+ CSCs by blocking the MAPK/ERK signaling pathway in HCC [38]. CSCs have become a research “hotspot” in recent years, and understanding the role of H19 in regulating CSCs will provide essential information for identifying new targets for HCC treatment.

### 3.7. Other Mechanisms

H19 has been reported to regulate angiogenesis by activating various signaling pathways or directly interacting with target proteins. Ischemia/reperfusion (I/R)-induced damage of HCC cells plays a beneficial role in the recovery of liver function. Chao Cui et al. reported that H19 induced hypoxia/reoxygenation injury by up-regulation of autophagy via activation of the PI3K-Akt-mTOR pathway in the HCC cells [40]. Jinqiang Zhang et al. developed a novel experimental system using tumor-initiating hepatocytes (TICs), which was used to demonstrate that TGF-β signaling in TICs inhibited H19 expression via Sox2. It is well established that the TGF-β/Sox2 signaling axis is crucial for inhibiting HCC development [44]. Recent studies in glioblastoma and bladder carcinoma confirm the interaction of H19 with enhancer of zeste homolog 2 (EZH2) regulation [90]. H19 has been shown to regulate the EZH2-Wnt/β-catenin signaling axis in HCC development [56,91]. Similarly, studies demonstrated the strong association between EZH2 and H19 to suppress Wnt antagonist genes to activate Wnt/β-catenin signaling in cancer development [92]. H19 overexpression directly reversed the suppressive Wnt/β-catenin signaling axis in HCC cells [57]. H19 was also reported as a pro-oncogenic during the development of chronic inflammation-mediated HCC in the Mdr2-KO mouse model, mainly by increasing liver injury and decreasing hepatocyte polyploidy in young mice [52]. The potential protein targets of H19 in HCC are shown in Figure 3.

## 4. LncRNA H19 in CCA

CCA is the second most malignant liver tumor after HCC. CCA originates mainly from cholangiocytes/bile duct cells [93]. It was historically a rare form of cancer but is becoming a growing threat to health due to its aggressiveness and poor prognosis [94]. CCA is often diagnosed at late stages with limited therapeutic options. Increasing evidence indicates that lncRNAs play critical roles in the carcinogenesis and development of CCA [95,96]. The expression of H19 is closely associated with cholangiocyte proliferation and senescence. Several studies have reported that the hepatic H19 expression is regulated by conjugated bile acids, and upregulation of H19 is associated with cholestatic liver injury in different mouse models of cholestasis as well as patients with primary biliary cholangitis (PBC) and primary sclerosing cholangitis (PSC) [97,98,99,100,101,102,103,104], indicating the potential role of H19 in CCA.

Bioinformatics analyses from microarray data showed that the expression of H19 is significantly upregulated in both CCA tissues and cell lines compared with nontumor tissues and normal cell lines, respectively [105]. Moreover, high H19 expression was significantly correlated with tumor size, advanced tumor node metastasis (TNM) stage, and postoperative recurrence of CCA [105]. Consistent with this result, a qualitative study revealed that H19 facilitated cell migration and invasion in CCA by targeting IL-6 via ceRNA patterns of sponging let-7a/let-7b [106]. In addition, suppression of H19 impaired CCA cell migration and invasion potential by reversing EMT [107]. It also has been reported that H19 enhanced the development of CCA by promoting the expression of Bcl-2 via inhibiting miR612 [108]. The most recent studies related to H19 in CCA are summarized in Table 3 and the potential mechanisms of H19 in CCA are illustrated in Figure 4. Studies with cultured CCA cell lines and human CCA samples indicated that H19 could promote the proliferation, migration, and invasion of CCA cells by acting as a miRNA sponge or affecting EMT. More mechanistic and comprehensive studies are needed to define the role of H19 in the progression of CCA.

## 5. Conclusions and Perspectives

Accumulating evidence demonstrated the critical role of lncRNAH19 in the development of liver cancer, including HCC and CCA. H19 is implicated in various pathological processes of cancer development, including cell proliferation, apoptosis, and EMT. H19 regulates gene expression at epigenetic, transcriptional, and post-transcriptional levels. The current understanding of H19 in liver cancer remains limited due to a lack of proper *in vitro* cell culture models and *in vivo* animal models as well as the small sample size of clinical studies. Recent advances in new technologies, such as CRISPR/Cas9-based gene editing, organoid cultures, single cell RNAseq, and GeoMx^®®^ Digital Spatial Profiler, will provide better opportunities to study the pathogenesis and progression of various cancers, including HCC and CCA. In summary, lncRNA H19 represents a promising diagnosis and prognosis biomarker for liver cancer as well as a promising therapeutic target.

## Figures and Tables

**Figure 1 ncrna-09-00019-f001:**
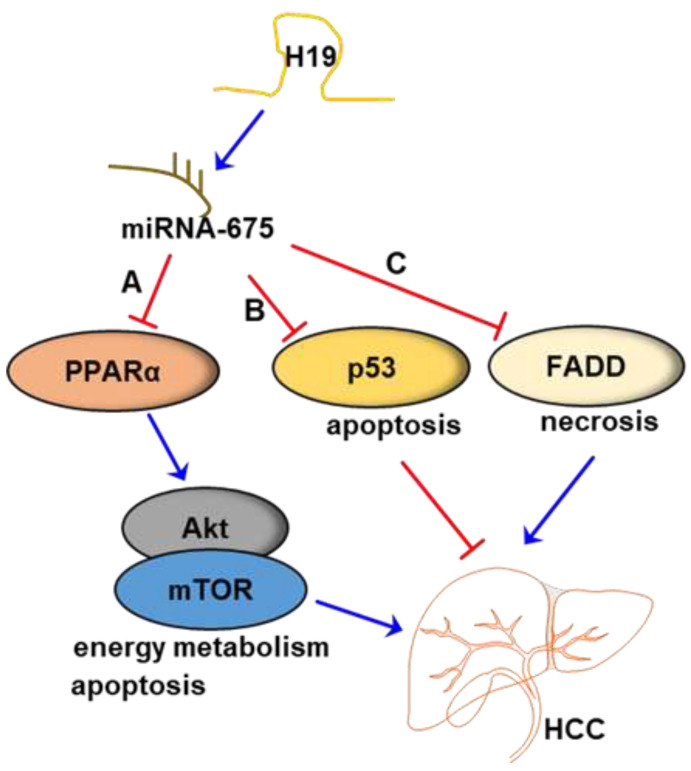
The potential mechanisms of the LncRNA H19/miR-675 axis in HCC. (A) H19/miR-675 inhibits the expression of PPARα, which further activates the Akt/mTOR signaling pathway, thus modulating energy metabolism and cell apoptosis, eventually promoting the development of HCC. (B) H19/miR-675 decreased the expression of p53, thus degrading cell apoptosis, eventually promoting the development of HCC. (C) H19 /miR-675 inhibited the expression of Fas-associated protein with death domain (FADD), thus reducing necrosis, eventually suppressing the development of HCC.

**Figure 2 ncrna-09-00019-f002:**
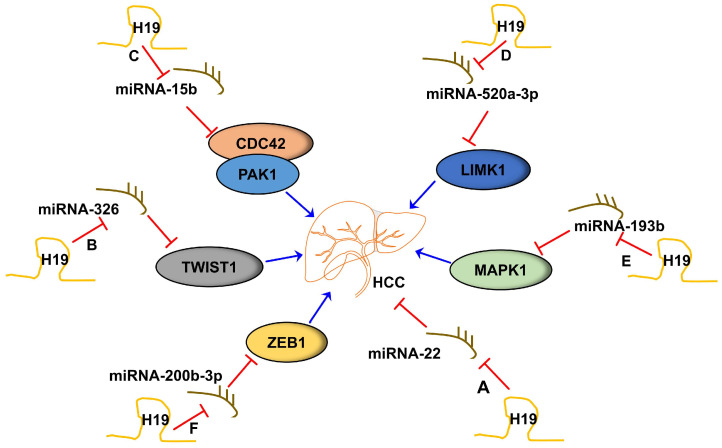
LncRNA H19 functions as miRNA sponges in HCC. (A) LncRNA H19 sponged miRNA-22, promoting the development of HCC. (B) LncRNA H19 sponged miR-326, thus increasing the expression of TWIST1, eventually promoting the development of HCC. (C) LncRNA H19 sponged miR-15b, thus activating the CDC42/PAK1 signaling pathway, promoting the development of HCC. (D) LncRNA H19 sponged miR-520a-3p, thus increasing the expression of LIMK1, eventually promoting the development of HCC. (E) LncRNA H19 sponged miR-193b, thus increasing the expression of MAPK1, eventually promoting the development of HCC. (F) LncRNA H19 sponged miR-200b-3p, thus upregulating the expression of zinc finger E-box binding homeobox 1 (ZEB1), eventually promoting the development of HCC.

**Figure 3 ncrna-09-00019-f003:**
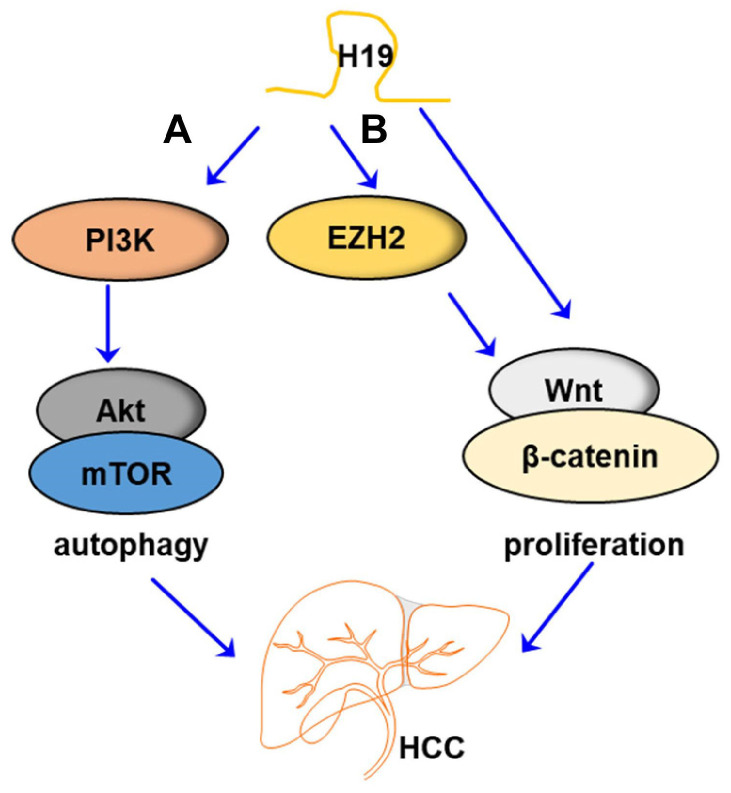
The potential targets of LncRNA H19 in HCC. (A) H19 induced the expression of PI3K, which further activated the Akt/mTOR signaling pathway, thus modulating autophagy and eventually promoting the development of HCC. (B) H19 induced the expression of EZH2, which further activated the Wnt/β-catenin signaling pathway, thus regulating proliferation and eventually promoting the development of HCC.

**Figure 4 ncrna-09-00019-f004:**
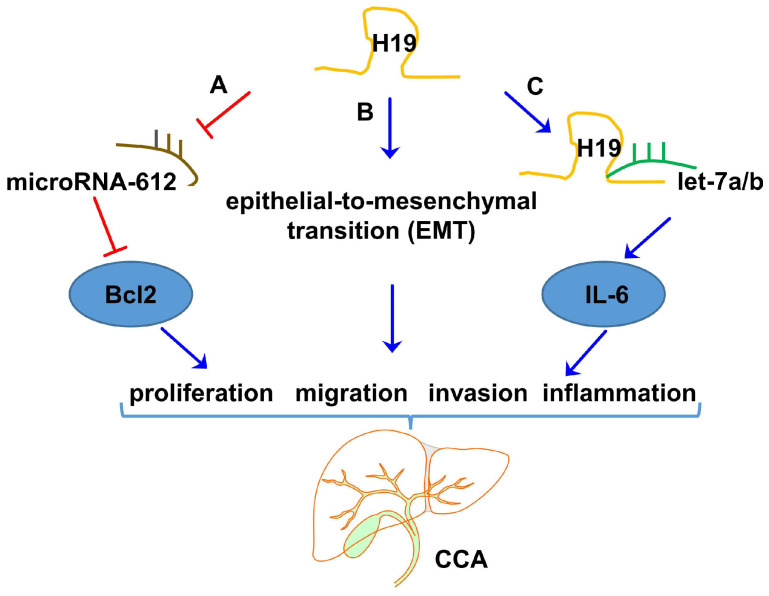
Potential mechanisms of LncRNA H19 in CCA. (A) LncRNA H19 promoted the expression of Bcl-2 by downregulating the expression of miR-612, thus promoting the proliferation, migration, and invasion of cholangiocarcinoma cells. (B) LncRNA H19 promoted cell migration and invasion by affecting EMT and resulted in a poor prognosis for cholangiocarcinoma. (C) LncRNA H19 sponged let-7a/b, thus increasing the expression of inflammation-related genes, including IL-6, resulting in abnormal inflammation responses and pathogenesis of CCA.

**Table 1 ncrna-09-00019-t001:** Potential targets of lncRNA H19 in HCC.

Animal Models or Human Samples	*In Vitro* Models	Targets	Potential Mechanisms	Reference and Year
HCC patients and cancer-free controls (*n* = 42)	Huh 7	oxidative stress MAPK/ERK	Downregulation of H19-induced oxidative stress and reversed chemotherapy resistance of CD133+ cancer stem cells by blocking the MAPK/ERK signaling pathway in HCC	[38], 2018
HCC patients (*n* = 214)	HepG2 and huh-7 cell lines	miR-675/FADD/caspase-8/caspase-3	H19 upregulated the expression of miR-675, reducing the expression of FADD, caspase-3, and caspase-8.	[39], 2018
_	HepG2 and HCCLM3 cell lines	PI3K-Akt-mTOR/ autophagy	H19 induced the expression of PI3K, which further activated the Akt/mTOR signaling pathway, thus modulating autophagy in HCC	[40], 2019
HBV-related HCC liver tissues and matched normal tissue (*n* = 20)	HepG2, HepG2.2.15, and LO2 cell lines	miRNA-22	H19 promoted the development of hepatitis B-related HCC by regulating miRNA-22 via the epithelial-mesenchymal transition (EMT) pathway	[41], 2019
HBV patient liver tissues and matched normal tissue (*n* = 20)	LO2 cell lines	miR-675/PPARα Akt/mTOR	H19/miR-675 suppressed the expression of PPARα, which further activated the Akt/mTOR signaling pathway, thus modulating energy metabolism and cell apoptosis	[42], 2019
_	Hep3B, HepG2, MHCC-97L, SK-hep1, Hun7, SMCC-7721 and LO2 cell lines	miR-326/TWIST1	H19 sponged miR-326, thus increasing the expression of TWIST1, eventually promoting the development of HCC	[43], 2019
C57/BL6J mice transplanted with TICs from DEN-treated Tgfbr2^fl/fl^ mice by splenic injection followed by i.p. injection of CCl4 and tail vein injection of Ad-Cre.	TICs isolated from B6.129S6-Tgfbr2^fl/fl^ mice injected with DEN	TGFβ/Tgfbr2-Sox2	TGF-β signaling in TICs inhibited H19 expression via Sox2, which was crucial for the inhibition of HCC development	[44], 2019
HCC patient liver tissues and the adjacent normal liver tissues (*n* = 46)	HepG2, SMMC-7721, Bel-7402, and Huh-7 cell lines	miR-15b/CDC42/PAK1 axis	H19 activated CDC42/PAK1 pathway to promote cell proliferation, migration, and invasion by targeting miR-15b in hepatocellular carcinoma	[45], 2019
HCC patient liver tissues and matched noncancerous liver tissues (*n* = 55)	HepG2 cell lines	NSUN2 G3BP1/Myc	NSUN2-mediated m5C-modified H19 promoted HCC by recruiting G3BP1 oncoprotein, which leads to MYC accumulation	[46], 2020
BALB/c nude mice injected with Huh7 cells, then injected with exosomes derived from Huh7 cells treated with Propofol and Vector or Over H19.	Huh7, MHCC97-H, and HCCLM3 cell lines	miR-520a-3p/ LIMK1 axis	Exosomal H19 facilitated the malignant potential of Propofol-exposed HCC cells via miR-520a-3p/LIMK1 axis	[47], 2020
HCC patient liver tissues (*n* = 242) with matched nontumor tissues (*n* = 298)	Huh7, Hep3B, SNU-449, and SNU-387 cell lines	miR-675	Knockdown of H19 sensitized HCC cells to sorafenib by downregulating miR-675, thereby preventing EMT	[48], 2020
HCC patient liver tissues (*n* = 64) and TCGA cohort (*n* = 393)	HepG2, Hep2B2, THP-1, SK-OV-3, and NCI-H520 cell lines	miRNA-193b MAPK1 axis	TAM-derived H19 promoted tumor cell migration and invasion and immune cell infiltration by hijacking miR-193b as a sponge and activating MAPK	[49], 2020
Srsf2^f/f^ mice and Srsf2^f/f^-Mx1cre mice	_	Srsf2/IGF2 PI3K/Akt MAPK/ERK	Demethylation-induced high expression of IGF2/H19, followed by activation of PI3K/Akt and MAPK/ERK signaling, contributed to the tumorigenesis of Srsf2 HKO mice	[50], 2020
Nude mice injected with pcDNA3.1-H19, and H19-KO cells or control	MHCC97H, MHCC97L, and HCC-LM3 cell lines	p53	Knockdown of H19 induced the protein expression of p53, eventually promoting cell apoptosis	[51], 2020
16-month-old female and 17.7-month-old male C57/BL6 Mdr2^−/−^ and Mdr2^−/−^/H19^−/−^ DKO mice	Primary cells	liver injury/hepatocyte polyploidy	H19 was a pro-oncogenic during the development of chronic inflammation-mediated HCC by increasing liver injury and decreasing hepatocyte polyploidy	[52], 2020
HCC patient samples and matched healthy controls (*n* = 60); BALB/C mice injected with miR-675 or siCtrl	Huh7, HepG2, Hep3B, and FLC4 cell lines	miR-675/FADD	miR-675 repressed FADD, leading to the development of necroptosis	[53], 2021
HCC patient samples (*n* = 81) BALB/cA-nude mice injected with BM4-1/H19 and BM4-1/H19	MDA-MB-231 and PC-3 cell lines	PPP1CA/p38MAPK and miR-200b-3p	H19 promoted HCC bone metastasis by upregulating zinc finger E-box binding homeobox 1 (ZEB1) via functioning as a sponge for miR-200b-3p	[54], 2021
HCC patients and matched healthy controls (*n* = 42)	BEL-7402, Huh-7, and HL-7702 cell lines	miR-140-5p/EMT process and apoptosis.	H19 targeted miR-140-5p and effectively inhibited the EMT process of HCC cells and promoted apoptosis	[55], 2021
_	Hep3B and SMMC-7721 cell lines	EZH2/ Wnt/β-catenin	EZH2 interacted with H19 in HCC development in regulating Wnt/β-catenin signaling	[56], 2021
_	LO2, SMMC-7721 and HepG2 cell lines	Wnt/β-catenin	H19 overexpression activated canonical Wnt/β-catenin signaling	[57], 2021
HCC patient tissues and matched healthy controls (*n* = 10)	Huh-7 cell lines	miR-186/IGF2BP1	miR-186 decreased IGF2BP1, thus inhibiting H19, exerting tumor suppressor effects in HCC	[58], 2022

Note: Problematic cell line: LO2, contaminated, may be a HeLa derivative.

**Table 2 ncrna-09-00019-t002:** H19 polymorphisms and their association with HCC risk.

Number of Cases	H19 Variant Associated with HCC Risk	Reference and Year
472 HCC patients and 472 healthy controls	rs2839698	[61], 2018
214 HCC patients	rs217727	[39], 2018
359 HCC patients and 1190 healthy controls	rs2839698 rs3741219 rs2107425 rs3024270	[62], 2019
273 human HCC patients treated with transarterial chemoembolization and 26 HCC patients receiving curative resection	rs3741219	[63], 2021

**Table 3 ncrna-09-00019-t003:** Potential targets of lncRNA H19 in CCA.

Animal Models or Human Samples	*In Vitro* Models	Targets	Potential Mechanisms	Reference and Year
year_	QBC939, SK-cha-1, and RBE cell lines	let-7a/IL-6	H19 functioned as competing endogenous RNAs by sponging let-7a/b, which activated pivotal inflammation cytokine IL-6.	[106], 2016
Human CCA tissues and corresponding adjacent non-tumor tissues (*n* = 56)	QBC939 and RBE cell lines	EMT	H19 promoted cell migration and invasion by affecting EMT	[107], 2017
Human CCA tissues and matched normal bile duct tissues (*n* = 43)	HUCCT1, QBC939, HCCC 9810, and RBE cell lines	HIF1α/miRNA-612/Bcl-2 axis	Transcription factor HIF1α promoted proliferation, migration, and invasion of cholangiocarcinoma via H19/miRNA-612/Bcl-2 axis	[108], 2020

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
