# Peer review of "Long Noncoding RNA H19: A Novel Oncogene in Liver Cancer"

_ncrna, 2023, doi:10.3390/ncrna9020019_

Round 1

Reviewer 1 Report

In this paper, they reviewed the role of a long non-coding RNA, H19, in liver cancer. H19 is a lncRNA, which has been extensively studied. Multiple studies have demonstrated the correlation between H19 and HCC and tried to explore the mechanisms. They have summarized those studies in a systematic way. They also reviewed the mechanisms of H19 proposed by those studies.

Overall, this review article is well written and provide a comprehensive overview for researcher studies H19 and HCC.

Author Response

We would like to thank this reviewer for his or/her time to review our manuscript and positive comments.  We checked the manuscript for grammar and spelling errors.

Reviewer 2 Report

In this review, Y. Wang, et.al. summarized the action mechanisms and the regulatory roles of H19 in liver cancers. It was well-prepared and written and comprehensive. There are some points that need to be addressed:

1.       Line 50, why is H19 not suppressed in skeletal muscles (cartilage and cardiac muscle) as well? Do the authors have any thoughts about that?

2.       An illustration of H19 and IGF2 gene expression may further benefit the readers in understanding how H19 is expressed and epigenetically controlled.

3.       The “LO2 cell line” is misidentified; should be pointed out in the context.

4.       In Fig.1, “Akt/mTOR signaling pathway, thus modulating energy metabolism and cell apoptosis, eventually promoting the development of HCC”. Does AKT/mTOR promote HCC? It looks like that the authors used a suppressive indicator in the figure. Please clarify.

5.       How is H19 expressed in the pre-cancerous human cirrhotic livers?  

6.        Does H19 regulate innate immunity in the pathogenesis of HCC/CCA?

Author Response

We want to thank this reviewer for his or /her careful review and constructive comments. We revised the manuscript based on the comments. The following is the point-to-pint response for each comment.

  1. Line 50, why is H19 not suppressed in skeletal muscles (cartilage and cardiac muscle) as well? Do the authors have any thoughts about that?

Response: Thank you for mentioning this. The reason why H19 is not suppressed in muscles in adults is not clear. One possibility is that the specific transcription factors and signaling pathways to promote H19 expression are present in muscle cells and may be absent or less active in other tissues. Recent studies have reported that H19 can regulate the proliferation and differentiation of muscle cells [1-4].

  1. An illustration of H19 and IGF2 gene expression may further benefit the readers in understanding how H19 is expressed and epigenetically controlled.

Response: Thanks for the suggestion. The regulation of H19 and IGF2 expression was illustrated in our previous review [5].

  1. The “LO2 cell line” is misidentified; should be pointed out in the context.

Response: Thanks for the suggestion. The "LO2 cell line" has been found to be misidentified or contaminated with other cell lines in several studies. The misidentification or contamination of cell lines can lead to incorrect experimental results and hinder scientific progress. The contamination of LO2 cells has been noted in the relevant tables as “Problematic cell line: LO2, contaminated, maybe a HeLa derivative”.

  1. In Fig.1, “Akt/mTOR signaling pathway, thus modulating energy metabolism and cell apoptosis, eventually promoting the development of HCC”. Does AKT/mTOR promote HCC? It looks like that the authors used a suppressive indicator in the figure. Please clarify.

Response:  Thanks for pointing this out. The activation of the Akt/mTOR signaling pathway promotes the malignant progression of HCC, so Akt/mTOR should be depicted as an inductive action in Fig. 1. The figure has been revised accordingly.

  1. How is H19 expressed in the pre-cancerous human cirrhotic livers?  

Response: In human samples, H19 expression was higher in females and positively correlated with liver cirrhosis in non-tumor liver samples [6]. Besides, our previous study also showed that serum exosomal-H19 level was gradually upregulated during cholestatic disease progression [7]. These studies suggest that H19 is upregulated in pre-cancerous human cirrhotic livers, and its expression may be positively correlated with liver cirrhosis progression. The exact mechanism of how H19 contributes to the pathogenesis of liver cirrhosis remains unclear and requires further investigation.

  1. Does H19 regulate innate immunity in the pathogenesis of HCC/CCA?

Response:  At present, there is limited direct evidence that H19 regulates innate immunity in the pathogenesis of HCC/CCA. However, several studies suggest a potential role of H19 in regulating innate immunity in these diseases. It has been shown that tumor-associated macrophages (TAMs)-derived H19 promoted HCC tumor cell migration, invasion, and immune cell infiltration by sequestering miR-193b as a sponge and activating MAPK signaling [8]. Additionally, H19 was reported to regulate NLRP3 in other tissues [9,10]. Furthermore, H19 was found to regulate the expression of IL-10, an anti-inflammatory cytokine, in human monocytes. IL-10 plays a crucial role in regulating the innate immune response and has been reported to be involved in the pathogenesis of HCC. Taken together, these studies suggest that H19 may have a potential role in regulating innate immunity in the pathogenesis of HCC/CCA, and further studies are needed.

References

  1. Kumar, A.; Datta, M. H19 inhibition increases HDAC6 and regulates IRS1 levels and insulin signaling in the skeletal muscle during diabetes. Molecular Medicine. 2022, 28, 1-14.
  2. Gabory, A.; Jammes, H.; Dandolo, L. The H19 locus: role of an imprinted non‐coding RNA in growth and development. Bioessays. 2010, 32, 473-480.
  3. Dey, B.K.; Pfeifer, K.; Dutta, A. The H19 long noncoding RNA gives rise to microRNAs miR-675-3p and miR-675-5p to promote skeletal muscle differentiation and regeneration. Genes & development. 2014, 28, 491-501.
  4. Geng, T.; Liu, Y.; Xu, Y.; Jiang, Y.; Zhang, N.; Wang, Z.; Carmichael, G.G.; Taylor, H.S.; Li, D.; Huang, Y. H19 lncRNA promotes skeletal muscle insulin sensitivity in part by targeting AMPK. Diabetes. 2018, 67, 2183-2198.
  5. Wang, Y.; Hylemon, P.B.; Zhou, H. Long Noncoding RNA H19: A Key Player in Liver Diseases. Hepatology. 2021, 74, 1652-1659.
  6. Gamaev, L.; Mizrahi, L.; Friehmann, T.; Rosenberg, N.; Pappo, O.; Olam, D.; Zeira, E.; Bahar Halpern, K.; Caruso, S.; Zucman-Rossi, J. The pro-oncogenic effect of the lncRNA H19 in the development of chronic inflammation-mediated hepatocellular carcinoma. Oncogene. 2021, 40, 127-139.
  7. Li, X.; Liu, R.; Huang, Z.; Gurley, E.C.; Wang, X.; Wang, J.; He, H.; Yang, H.; Lai, G.; Zhang, L. Cholangiocyte‐derived exosomal long noncoding RNA H19 promotes cholestatic liver injury in mouse and humans. Hepatology. 2018, 68, 599-615.
  8. Ye, Y.; Guo, J.; Xiao, P.; Ning, J.; Zhang, R.; Liu, P.; Yu, W.; Xu, L.; Zhao, Y.; Yu, J. Macrophages-induced long noncoding RNA H19 up-regulation triggers and activates the miR-193b/MAPK1 axis and promotes cell aggressiveness in hepatocellular carcinoma. Cancer letters. 2020, 469, 310-322.
  9. Yang, H.; Zhang, Y.; Du, Z.; Wu, T.; Yang, C. Hair follicle mesenchymal stem cell exosomal lncRNA H19 inhibited NLRP3 pyroptosis to promote diabetic mouse skin wound healing. Aging. 2023, 15.
  10. Liu, Y.; Luo, Y.; Zhang, A.; Wang, Z.; Wang, X.; Yu, Q.; Zhang, Z.; Zhu, Z.; Wang, K.; Chen, L. Long Non-coding RNA H19 Promotes NLRP3-Mediated Pyroptosis After Subarachnoid Hemorrhage in Rats. Translational Stroke Research. 2022, 1-15.